# Flexor tendon degeneration affects short-term outcomes of open trigger digit release

**Mardhiyah Abdul Nasir**[1⦿]**, Tunku Sara Ahmad**[1⦿]**, Tze Hau Low**[2⦿]**, Cassidy Devarajooh**[3⦿]**, Jayaletchumi Gunasagaran**[1]*

1 National Orthopaedic Centre of Excellence for Research and Learning (NOCERAL), Department of Orthopaedic Surgery, Faculty of Medicine, Universiti Malaya, Kuala Lumpur, Malaysia, 2 Subang Jaya Medical Centre, Subang Jaya, Selangor, Malaysia, 3 Bentong District Health Office, Ministry of Health, Bentong, Pahang, Malaysia

⦿ These authors contributed equally to this work.
* jayaletchumi@um.edu.my

**Data Availability Statement:** All relevant data are within the paper and its Supporting Information files.

**Funding:** The author(s) received no specific funding for this work.

## Abstract

We aimed to investigate the association between flexor tendon degeneration and outcome of open trigger digit release. We recruited 162 trigger digits (136 patients) who had open trigger digit release from February 2017 to March 2019. Intraoperatively, six features of tendon degenerations were identified: irregular tendon surface, tendon fraying, intertendinous tear, synovial thickening, hyperaemia of sheath and tendon dryness. Longer duration of preoperative symptoms was associated with worsening tendon surface irregularity and fraying; increased number of steroid injections was associated with worsening tendon surface irregularity and dryness; higher DASH score was associated with severe tendon fraying, dryness and intertendinous tear; limited proximal interphalangeal joint (PIPJ) motion was associated with severe tendon dryness. At 1-month post-surgery, DASH score remained high in severe intertendinous tear group while PIPJ motion remained limited in severe tendon dryness group. In conclusion, the severity of various flexor tendon degenerations influenced the outcome of open trigger digit release at 1-month but did not affect the outcome at 3- and 6-months post-surgery.

## Introduction

Trigger finger is caused by pathologic discrepancy between flexor tendon diameter and the diameter of fibro-osseous canal formed by A1 pulley [1, 2] and rarely with involvement of proximal part of A2 pulley [3]. Gliding of flexor tendon in a narrow-restricted sheath further predisposes the inner layer of A1 pulley to hypertrophy and chondroid metaplasia, while the flexor tendons will inevitably undergo repeated mechanical compression [4, 5]. Repeated and forceful friction between flexor tendon and A1 pulley may cause the flexor tendon to degenerate. Most literature reviews have only focused on A1 pulley analysis rather than the flexor tendon itself, and this is probably due to its accessibility to histologic examination [6, 7]. A few

**Competing interests:** The authors have declared that no competing interests exist.

ultrasound imaging studies on flexor tendons in trigger finger revealed high incidence of flexor tendinopathy and it was associated with severity of the trigger finger [8–11]. Recent two studies by Lundin et al demonstrated tendinosis in all biopsied flexor digitorum superficialis tendons in trigger fingers [12, 13]. In open trigger finger release, macroscopic irregularity, fraying of tendon surfaces [14, 15] and laceration at the edge of flexor tendon, are common findings consistent with tendon degeneration. The purpose of our study was to investigate the correlation between flexor tendon degeneration and the outcome of open A1 pulley release in patients with trigger digit.

## Methods

The research was conducted in accordance with ethical principles defined in the Declaration of Helsinki and Good Clinical Practice Guideline. Ethical approval for this research was obtained from Medical Research Ethics Committee.

### Sample size

Sample size was calculated by using the Sample Size Calculator for Estimations [16]. The prevalence of flexor tendon degeneration was set as 22% and 62% respectively [9, 17] to determine the minimal sample size required. The sample size calculated was 132 digits. In our study, 162 digits were included.

### Patients

This prospective study included 136 patients (88 females, 48 males; 162 digits) from February 2017 to March 2019 at a tertiary medical centre. Written informed consents were taken from the patients before initiating the study. Inclusion criteria were patients; age more than 18 years old and above with diagnosis of trigger digit requiring open trigger digit release. Exclusion criteria included pre-existing hand deformity, neurological disorder, previous hand injury, and inflammatory arthritis.

### Preoperative examination

Demographic and clinical data were obtained prior to surgery from patients at the minor operating theatre by orthopaedic registrars assisted by a research assistant. We followed Green et al in grading of trigger digit (Table 1). Indications for surgery were trigger digit grade II with failed trial of physiotherapy or corticosteroid injections, grade III and grade IV. Initial preoperative assessments include; pain score (VAS), range of motions (ROM) of metacarpophalangeal joints (MCPJ), proximal interphalangeal joints (PIPJ) and distal interphalangeal joints (DIPJ) using JAMAR® hand goniometer, grade of trigger digit, grip strength using JAMAR® Hand Grip Dynamometer, pinch strength using JAMAR® Hydraulic Pinch Gauge and a patient-based questionnaire using the English version of Disabilities of Arm, Shoulder, and Hand questionnaire (DASH) [18]. For the assessment of grip and pinch strengths, patients were seated straight on a chair with both feet placed flat on ground; shoulder adducted and neutrally rotated, elbow flexed 90˚, forearm and wrist in neutral position. The JAMAR® Hand Grip Dynamometer was fixed at handle position 2 for all patients. Tip-to-tip pinch strength of the affected digit was measured. Both grip and pinch strengths were measured 3 times; each time patients were instructed to use maximum power for 3 seconds. The maximum value was documented.

**Table 1. Green's classification for grading of the severity of trigger digit.**

| Grade | Clinical Findings |
|---|---|
| I (Pretriggering) | Pain; tenderness over A1 pulley; reported history of catching |
| II (Active) | Demonstrable catching on physical examination with preserved active extension |
| III (Passive) | IIIA: Catching requiring passive extension to release |
|  | IIIB: Loss of active flexion |
| IV (Contracture) | Fixed flexion contracture at the PIPJ |

Source: Wolfe SW: Tendinopathy, in Wolfe SW, Hotchkiss RN, Pederson WC, Kozin SH, Cohen MS, eds.: Green's Operative Hand Surgery, 7th Ed. Philadelphia: Elsevier, 2017; pp. 1903–1925.

## Intraoperative examination

Open surgical release of A1 pulley was done under local anesthesia with usage of pneumatic tourniquet around upper arm at a mean pressure of 250mmHg. All surgeries were performed by two levels of expertise; Orthopedics hand surgeons and registrars. Surface landmark of A1 pulley was initially delineated in the thumb and fingers, following Patel et al and Fiorini et al technique [19, 20]. About 15 mm transverse incision was placed at the level of marking of the surface anatomical landmark. A1 pulley was then visualized and divided longitudinally. Flexor digitorum superficialis (FDS) and profundus (FDP) or flexor pollicis longus (FPL) tendons were then inspected macroscopically using surgical loupe with 4.5x magnification (Carl Zeiss EyeMag® Pro S Medical Loupe, Carl Zeiss Meditec AG, Germany). The tendons were analyzed when the surgeons mildly tract both tendons out using a curved mosquito artery forceps or McDonald elevator. The traction was enough only to visualize 20mm of the tendons. The close-up image of the FDS or FPL tendon appearance was then captured using a 16.0-megapixel camera with microscopic mode and 4.0x focus at a distance about 100 mm from surgical field. Complete A1 pulley release was verified with tendon traction, active and passive range of motion of affected finger and no residual triggering. Proximal A2 pulley release was not performed as all our patients had complete resolution of triggering after A1 pulley release. The skin was closed with interrupted non-absorbable monofilament suture 4–0.

A panel of experts, involving five hand and microsurgeons (5–30 years of experience in hand fraternity) determined the features of flexor tendon degeneration which can be detected via intraoperative imaging. Six types of tendon degeneration features were identified; irregular tendon surface, tendon fraying, intertendinous tear or laceration, hyperaemia of tendon sheath, synovial sheath thickening and tendon dryness (Fig 1). A Likert scale between 1 to 5 was developed to grade the severity of each tendon degeneration feature; score 1: mild, score 3: moderate, score 5: severe. Score 0 was given if there was not any feature of tendon degeneration. Intraoperative imaging of the FDS or FPL tendon was then analyzed by two independent hand surgeons separately. Training was done for both surgeons prior to the assessment so they could identify and quantify the severity of tendon degeneration features (Likert scale 5) in a reliable and consistent manner. In this study, the scores obtained were averaged. Patients were discharged with advice of wound care and removal of sutures at day 14. Postoperative care instructions given included tendon gliding exercises. Patients were advocated to start the exercises at day 1 post-surgery. Compliance charts were given.

## Postoperative examination

Patients were required to attend postoperative follow ups on first, third and sixth month. During these visits, assessments were done at Upper Limb and Reconstructive Microsurgery Clinic

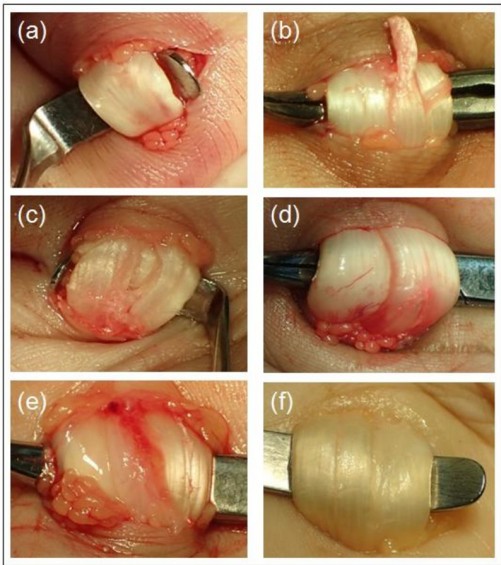

**Fig 1. Macroscopic appearances of flexor tendon degeneration.** (a) irregular tendon surface, (b) tendon fraying, (c) intertendinous tear or laceration, (d) hyperaemia of tendon sheath, (e) synovial sheath thickening, (f) tendon dryness.

by orthopaedics registrars, assisted by a research assistant and an occupational therapist. The preoperative and intraoperative assessments data were blinded from the knowledge of the assessors. Data obtained included pain score, ROM of MCPJ, PIPJ, and DIPJ, grip strength, pinch strength, presence of triggering symptom and DASH score (English version). Compliance towards post-surgery physiotherapy was checked.

**Table 2. Demographic and clinical data of total patients (N = 136) with trigger digit(s) who had open trigger digit release.**

| Variable | N (%) or Mean (SD) |
|---|---|
| **Gender** | |
| Male | 48 (35) |
| Female | 88 (65) |
| **Age** | 62 (SD 9.9) |
| **BMI** | 25.5 (SD 4.2) |
| **Associated comorbidities** | |
| No comorbid | 31 (23) |
| 1 comorbid | 27 (20) |
| 2 comorbids | 40 (30) |
| 3 comorbids | 31 (23) |
| 4 or more | 7 (6) |
| **Hand dominance** | |
| Right | 128 (94) |
| Left | 8 (6) |

SD, standard deviation; BMI, body mass index.

## Statistical analysis

Results of the continuous variables were described with mean and standard deviation and results of categorical variables were described with frequency and percentage. Pearson's correlation coefficient was used to evaluate and measure strength of association between severity of tendon degeneration with each outcome. In all statistical analyses, *p-value*< 0.05 was considered to be significant.

## Results

A total of 162 digits (136 patients) were initially recruited in this study. In the final analysis, 142 digits (120 patients) were assessed at 1-, 3- and 6-months postoperatively. Out of 162 digits recruited, 49 flexor tendons were normal and 113 were abnormal with presence of tendon degeneration. The demographic and clinical data of the patients with trigger digit are as listed in Tables 2 and 3. Middle finger was found to be the most affected with total number of 59 (36%), whereas little fingers were involved the least (1.8%). Right and left involvements were equally distributed in all digits. More than 1 digit were affected in 23 patients (17%).

Thirty-one patients (22.8%) had no comorbid, however, majority had multiple comorbidities including hypertension (56.5%), diabetes mellitus (47.1%), dyslipidaemia (53.7%), and ischemic heart disease (10.3%). Comorbid is defined as simultaneously existing medical conditions diagnosed by a medical professional. Most patients were retirees with total number of 61 (44.9%), while others included 31 housewives (22.8%), 21 office workers (15.4%), 10 professional officers (7.4%), five unskilled workers (3.7%), four skilled workers (2.9%) and four manual labourers (2.9%).

Six flexor tendon degeneration features were described, and they were found in various combinations (Fig 2). Distribution of tendon degeneration features according to its severity are summarized in Fig 3. The severity of irregular tendon surface (r = 0.180; p = 0.022) and fraying (r = 0.169; p = 0.031) were significantly associated with longer duration of symptoms. There were no correlations between tendon degenerations with grade or severity of trigger digit and duration of physiotherapy.

**Table 3. Overview of all trigger digits (N = 162).**

| Variable | N (%) or Mean (SD) |
|---|---|
| **Trigger finger grading** | |
| Grade II | 96 (59) |
| Grade III | 54 (33) |
| Grade IV | 12 (7) |
| **Duration of symptoms (month)** | 10.9 (SD 9.42) |
| **Affected digits** | |
| Dominant hand | 93 (57) |
| Non-dominant hand | 69 (43) |
| **Preoperative physiotherapy** | |
| Yes | 90 (56) |
| No | 72 (44) |
| Sessions | 34.05 (SD 12.11) |
| **History of steroid injection** | |
| Number | 39 (24) |
| Same affected digit | 26 (67) |

SD, standard deviation

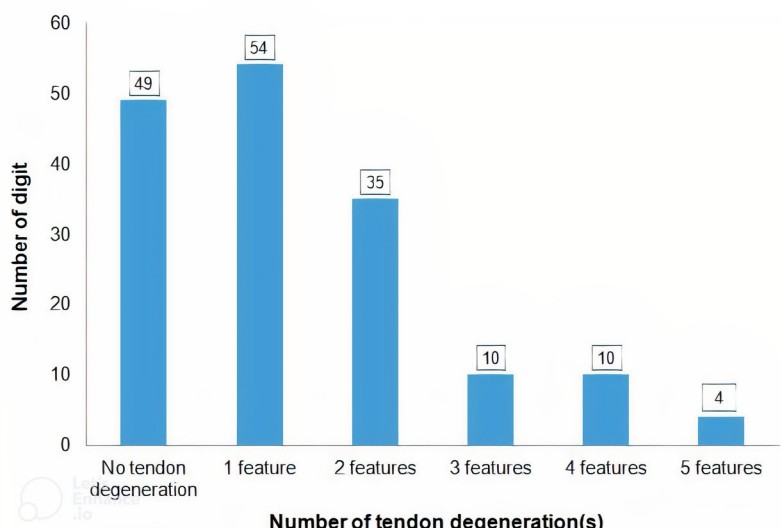

**Fig 2. Combination of tendon degeneration features.**

Interestingly, we found significant positive correlation between number of corticosteroid injections given at affected digits with severity of irregular tendon surface (r = 0.237; p = 0.002) and tendon dryness (r = 0.231; p = 0.003). Preoperative corticosteroid injections were administered into the tendon sheath in 26 digits. A maximum of three injections were given in two digits (grade II). Most of the injected trigger digits were of grade II (n = 17), few were grade III (n = 6) and grade IV (n = 3). The interval between injections and surgery was more than 6 months.

Preoperatively, higher DASH score was associated with severe tendon fraying, dryness and intertendinous tear; limited proximal interphalangeal joint (PIPJ) motion was associated with severe tendon dryness (Table 4). At 1-month post-surgery, DASH score remained high in severe intertendinous tear group while PIPJ motion remained limited in severe tendon dryness group (Table 4). At 3- and 6-months post-surgery, there were no significant differences between

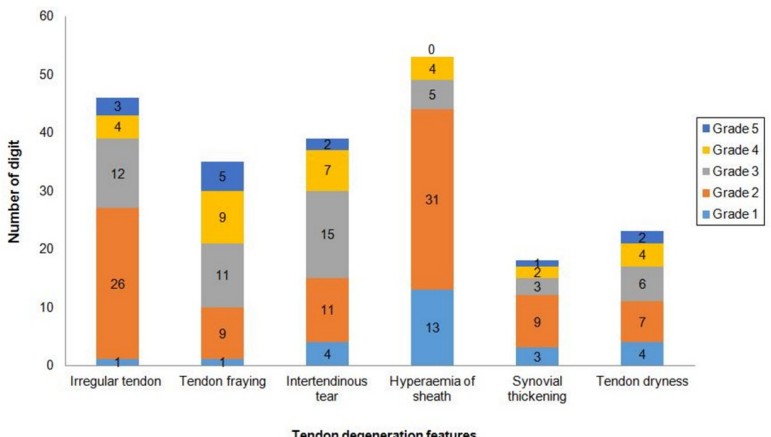

**Fig 3. Distribution of tendon degeneration features according to its severity.**

**Table 4. Correlation between tendon degeneration with number of steroid injections, preoperative and first month postoperative assessments.**

| | Number of previous steroid injections (N = 26) | Preoperative assessment (N = 162) | | | | | | First month postoperative assessment (N = 142) | | | | | |
|---|---|---|---|---|---|---|---|---|---|---|---|---|---|
| | | Pain score | ROM MCPJ | ROM PIPJ | Grip strength | Pinch strength | DASH score | Pain score | ROM MCPJ | ROM PIPJ | Grip strength | Pinch strength | DASH score |
| | r (p) | r (p) | r (p) | r (p) | r (p) | r (p) | r (p) | r (p) | r (p) | r (p) | r (p) | r (p) | r (p) |
| Irregular tendon | **0.237 (0.002)*** | -0.102 (0.198) | -0.013 (0.869) | 0.031 (0.692) | -0.092 (0.244) | 0.014 (0.863) | 0.056 (0.479) | -0.11 (0.194) | 0.096 (0.255) | 0.021 (0.808) | -0.061 (0.468) | 0.136 (0.106) | 0.038 (0.657) |
| Tendon fraying | 0.012 (0.878) | -0.075 (0.34) | -0.02 (0.768) | 0.047 (0.55) | -0.085 (0.283) | -0.04 (0.614) | **0.155 (0.049)*** | -0.179 (0.033) | 0.032 (0.706) | 0.032 (0.705) | -0.152 (0.071) | 0.056 (0.51) | 0.058 (0.493) |
| Intertendinous tear | 0.04 (0.611) | -0.044 (0.576) | -0.098 (0.214) | -0.141 (0.074) | 0.05 (0.527) | -0.036 (0.649) | **0.159 (0.044)*** | 0.016 (0.847) | -0.016 (0.849) | -0.07 (0.409) | 0.025 (0.769) | 0.114 (0.177) | **0.185 (0.028)*** |
| Hyperaemia of sheath | -0.015 (0.845) | 0.05 (0.524) | -0.011 (0.885) | 0.039 (0.619) | 0.126 (0.109) | 0.07 (0.377) | 0.01 (0.903) | -0.143 (0.089) | 0.049 (0.561) | 0.059 (0.484) | 0.073 (0.388) | 0.097 (0.249) | 0.025 (0.765) |
| Synovial thickening | -0.079 (0.317) | -0.078 (0.322) | 0.078 (0.325) | -0.047 (0.549) | -0.045 (0.573) | -0.132 (0.093) | 0.024 (0.758) | 0.049 (0.562) | 0.00 (0.998) | 0.018 (0.83) | -0.117 (0.165) | -0.075 (0.373) | 0.082 (0.33) |
| Tendon dryness | **0.231 (0.003)*** | -0.02 (0.799) | -0.013 (0.874) | **-0.183 (0.020)*** | -0.014 (0.86) | -0.081 (0.306) | **0.161 (0.041)*** | 0.028 (0.742) | -0.078 (0.359) | **-0.197 (0.018)*** | -0.075 (0.377) | 0.01 (0.903) | 0.106 (0.209) |

*Correlation is significant at <0.05 level (2-tailed).

ROM, range of motion; MCPJ, metacarpophalangeal joint; PIPJ, proximal interphalangeal joint; DASH, Disabilities of Arm, Shoulder, and Hand score.

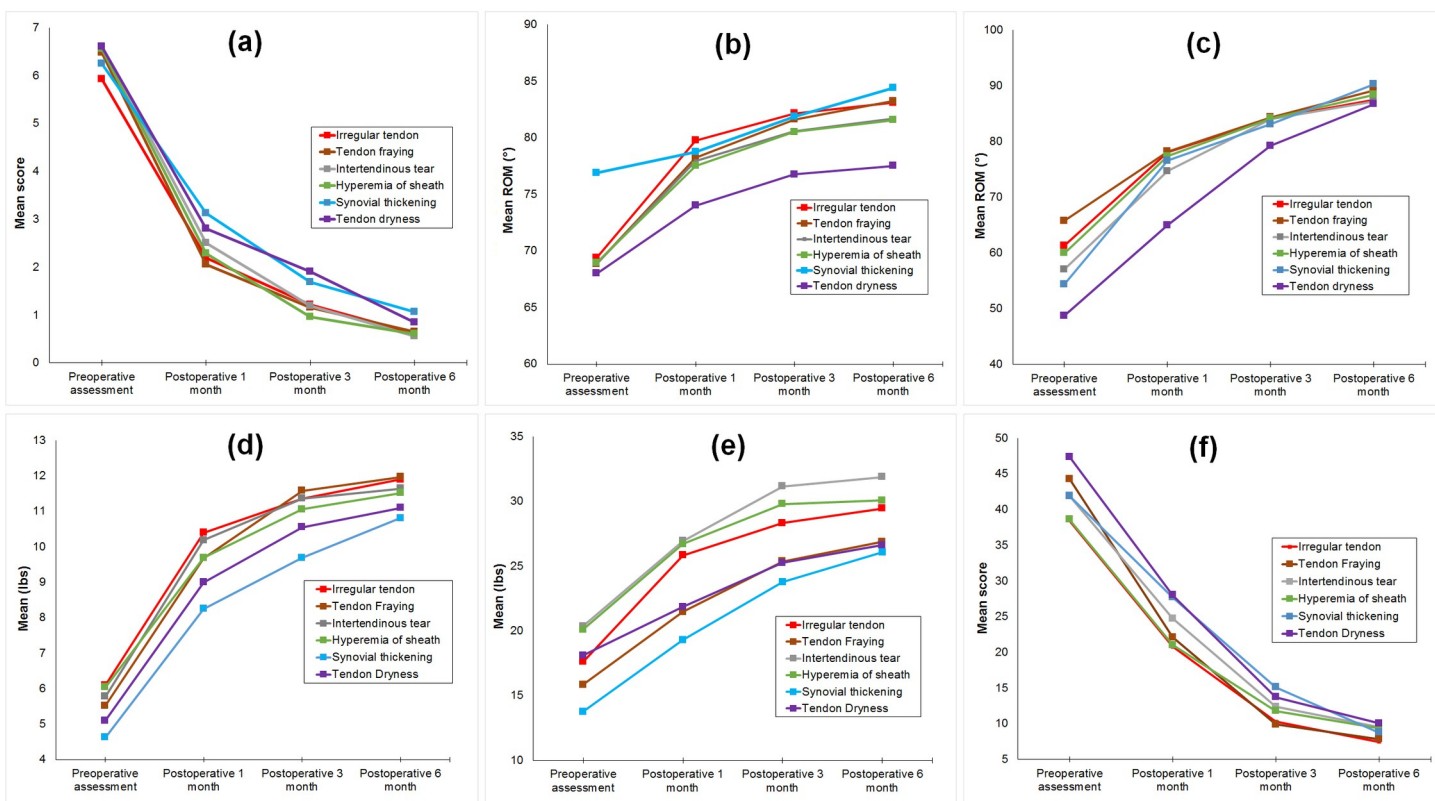

**Fig 4. Pre and postoperative clinical data according to different tendon degenerations.** (a) pain score, (b) ROM MCPJ, (c) ROM PIPJ, (d) pinch strength, (e) grip strength, (f) DASH score.

tendon degenerations and surgical outcome. There was significant improvement in all symptoms at 6 months after surgery regardless of type of degeneration in the tendons (Fig 4).

## Discussion

Flexor tendon degeneration is frequently seen intraoperatively but the incidence is under reported. Our study aimed to explore the role of flexor tendon degeneration in outcome of open trigger digit release. Open surgical release of trigger digit is currently a common practice in many centers all over the world after failed conservative management, due to its high success rate of up to 99% [21–23]. There were many literatures on short-term and long-term results of open trigger finger release, however, not many listed flexor tendon degeneration as a factor in the outcome.

In our study, flexor tendon degeneration incidence in trigger digit was high, as much as 70%, and this was comparable to Guerini et al. who reported 63% of flexor tendinosis or tenosynovitis sonographically in their series of 33 painful trigger fingers [8]. At present, ultrasonography and histological examination has been the main methods in identifying and describing flexor tendon degeneration in trigger digits.

Sonographically, all trigger fingers had thickened tendon [17], and a thickened hyperechogenic A1 pulley [8]. In addition, Guerini et al. reported 91% hypervascularization of A1 pulley, 48% flexor tendinosis, 55% tenosynovitis, 24% cystic appearance of synovial sheath (without tenosynovitis) and 6% intratendinous fissures. Combination of tendinopathies were also reported [8]. Serafini et al. investigated 66 tendons in trigger fingers and reported 15 cyst-like features, 20 synovial sheath thickening and 17 irregular internal echotextures. The two former features were more in Stage 1 and 2 trigger fingers whereas the tendon thickness and irregular internal echotextures increased with severity of trigger finger [17]. In a study by Kim et al involving 50 trigger digits, all the affected tendons were reported to be thickened, 14% loss of normal fibrillar echotexture, 62% blurring of tendon margin, 16% fluid collection in tendon sheath, 44% thickened A1 pulley, 6% sheath cysts and 4% of abnormal MCPJ [10]. Sato et al. described five abnormal features in trigger finger; segmental swelling of tendon, peritendinous hypoechoic rim, irregular internal echotexture, blurring of margin and hypervascularization of tendon sheath [11].

Histologically, Lundin et al. described microscopic features of tendon degeneration. They were micro-ruptures, collagen degradation, numerous cells with large round nuclei unevenly distributed in hyper- or hypocellularity, increased amount of ground substance and ingrowth of vessels. Some tendon specimens had presence of chondrocytes with surrounding hyalinization [12]. Uchihashi et al. found chondrocytoid cells producing hyaluronic acid and hypocellular collagen matrix in tenosynovium of trigger fingers [24]. However, there is scarce of literatures on macroscopic features of flexor tendon described in trigger digits. Lee et al reported 34% of flexor tendon fraying in their series of 26 trigger fingers and speculated that it could be the reason for persistent discomfort after open trigger release [15]. Baek et al. noticed 12 digits with flexor tendon injuries in their series of 109 trigger fingers and found it associated with prolonged postoperative symptoms [14]. However, both studies did not mention about other features of tendon degeneration. Thus, we attempted to identify and quantify the severity of flexor tendon degeneration via intraoperative imaging. Subsequently, we investigated the association of flexor tendon degeneration with outcome of open trigger digit release. In our knowledge, this is the first study to utilize intraoperative imaging of FDS and FPL tendons to assess the severity of flexor tendon degeneration in trigger digits.

We did not find any correlation between flexor tendon degeneration with severity of trigger digit at the point of presentation. This finding contradicted with Sato et al who stated that the

sonographic thickness of the flexor tendons under the A1 pulley was proportional to the severity of trigger finger [11]. We believe that the severity of trigger digit has multifactorial cause, and flexor tendinosis may contribute to a smaller percentage of the factors.

In our study, longer duration of symptoms was associated with severe flexor tendon fraying and irregular tendon surface. Initially, trigger digit may present as pain at the A1 pulley area, then subsequently into a painless clicking digit, which may then progress as a painful stiff digit, followed by loss of full range of motion [1]. The pathological transformation in A1 pulleys occurs due to an increase in pressure and abnormal frictional forces during finger movements [7]. Thus, the mechanical injuries on tendon surfaces probably worsen with time. Duration of preoperative symptoms and presence of tendon injuries were associated with prolonged postoperative symptoms [14]. Thus, patients should be advised not to delay their surgeries; if surgery is indicated.

Most literatures regarding tendon degeneration features, excluded patients who had preoperative corticosteroid injections. Only Lee J et al. reported that out of nine patients who had tendon fraying, four received corticosteroid injections [15]. We found significant correlation between number of corticosteroid injections and tendon degeneration; increased number of injections associated with worsening irregularity of tendon surface and dryness. This could be due to direct needle injury during the injection or effect of the steroid deposition in the tendon sheath [15]. Thus, extrasynovial injections localised at A1 pulley can be considered in order to reduce the risks of tendon degeneration [1].

We noticed that the severe intertendinous tear and tendon dryness were associated with high DASH score and limited ROM at PIPJ respectively at 1-month post-surgery. Management of these tendon degeneration features are not necessary as they were not associated with outcome at 3- and 6-months post-surgery. We believe that gradual healing; reparative process of degenerated flexor tendons occurs once the mechanical compression is removed. Kim et al reported that the tendon healing rate was significantly higher in patients who underwent rotator cuff repair and intraoperative application of thermosensitive antiadhesive gel [25]. We are unsure if intraoperative application of this agent may improve the intertendinous tear and tendon dryness which may improve the short-term outcome of open trigger digit release. Further study should be done to prove this hypothesis.

We initially hypothesized that pain will be more in tendinosis, as mentioned by earlier authors in their researches [26, 27]. Pain in tendinosis is due to biochemical and mechanical factors; including chemical irritants and collagen breakdown. In our study, all patients had significant improvement in pain at the affected digit after open trigger digit release. The release of A1 pulley halts the process of repetitive tendon microtrauma and healing, thus eliminating injury and pain. Similar finding had been reported by Scheller et al. after surgical release of DeQuervain's tenosynovitis [28].

In conclusion, duration of preoperative symptoms was associated with various flexor tendon degenerations. The severity of various flexor tendon degenerations influenced the outcome of open trigger digit release; high DASH score and limited PIPJ motion at 1-month but did not affect the outcome at 3- and 6-months post-surgery. Since it affects the short-term postoperative recovery, patients should be advised not to delay their surgeries if trigger digit release is indicated.

## Limitations

Even though the macroscopic assessment of flexor tendon degeneration maybe used to predict the short-term outcome of open trigger digit release, it must be cautioned that it is operator-dependent, and the accuracy maybe affected by surgeons' years of experience in the field.

## Supporting information

**S1 Data.**
(XLSX)

## Author Contributions

**Conceptualization:** Tze Hau Low, Jayaletchumi Gunasagaran.

**Data curation:** Mardhiyah Abdul Nasir, Cassidy Devarajooh.

**Formal analysis:** Cassidy Devarajooh.

**Investigation:** Tunku Sara Ahmad.

**Methodology:** Tze Hau Low, Jayaletchumi Gunasagaran.

**Resources:** Mardhiyah Abdul Nasir.

**Software:** Cassidy Devarajooh.

**Supervision:** Tunku Sara Ahmad.

**Validation:** Mardhiyah Abdul Nasir, Cassidy Devarajooh, Jayaletchumi Gunasagaran.

**Writing – original draft:** Mardhiyah Abdul Nasir.

**Writing – review & editing:** Tunku Sara Ahmad, Tze Hau Low, Jayaletchumi Gunasagaran.

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
