## [Decision Letter · Decision Letter 0]

12 Jan 2023

PONE-D-22-30862Flexor tendon degeneration affects short-term outcomes of open trigger finger releasePLOS ONE

Dear Dr. Gunasagaran,

Thank you for submitting your manuscript to PLOS ONE. After careful consideration, we feel that it has merit but does not fully meet PLOS ONE’s publication criteria as it currently stands. Therefore, we invite you to submit a revised version of the manuscript that addresses the points raised during the review process.

We look forward to receiving your revised manuscript.

Kind regards,

Lucinda Shen, MSc

Staff Editor

PLOS ONE

Journal Requirements:

Additional Editor Comments:

The manuscript has been reviewed by 2 reviewers and their comments may be seen below.

The reviewers have provided feedback on the manuscript, and have requested additional information to further clarify the study methodology. Specifically, they have raised questions about the number of surgeons who were involved in assessing the pictures used in the study. If multiple surgeons were involved, it would be beneficial for the readers to have more information on how any discrepancies were addressed, and what measures were taken to account for inter-rater variability. Additionally, the reviewers have suggested that providing more information on the Greens grading system used in the study would be useful for the readers' understanding and interpretation of the results.

Could you please carefully revise the manuscript to address all comments raised?

Reviewers' comments:

Reviewer's Responses to Questions

**Comments to the Author**

1. Is the manuscript technically sound, and do the data support the conclusions?

Reviewer #1: Yes

Reviewer #2: Yes

2. Has the statistical analysis been performed appropriately and rigorously? 

Reviewer #1: Yes

Reviewer #2: Yes

3. Have the authors made all data underlying the findings in their manuscript fully available?

Reviewer #1: No

Reviewer #2: Yes

4. Is the manuscript presented in an intelligible fashion and written in standard English?

Reviewer #1: No

Reviewer #2: Yes

5. Review Comments to the Author

Reviewer #1: The authors present an ambitious prospective study evaluating the effect of tendon degeneration on postoperative outcome following open trigger finger release.

The manuscript requires an English revision.

The introduction is short and on point.

In the methods section, please specify at what age you considered patients to be adults. I would also suggest to add the specifics of the grading system by Green that you used, as not all readers will be familiar with it.

Who performed the preoperative assessment and when was it performed? Did you use the English version or a translated version of the DASH?

When analysing the pictures, did the two surgeons do it together? Or separately? If separately, how did you handle the cases where the surgeons disagreed?

How did you define comorbidity?

The authors have included several digits from the same patients, and this is a recurrent problem in hand surgical research - how to handle multiple digits from the same patients. Perhaps a small elaboration on this is the discussion would add to the manuscript - for example, it is more likely that a patient with diabetes is represented by more than one finger in the data set than a patient who is otherwise healthy.

I would also suggest to add a regression analysis, to be able to adjust for confounding factors. Is the seen effect of tendon degeneration on ROM and DASH a true effect, or does age and comorbidity play a role?

In the results section, I was very surprised to see that patients had on average 34 visits to the physio before surgery, and few patients had recieved corticosteroid injections beforehand. This differs quite substantially from my practice. Is it standard in your practice that patients have such an extensive treatment by physios before surgery?

I am also curious of how you reason with corticosteroids, would a previous injection affect the appearance of the tendon?

In the discussion, it is mentioned that sonographic assessment was made, however this is not specified in the methods- or is line 210-211 only referring to previous literature?

Figure 4 is difficult to read and requires a higher pixel solution.

Reviewer #2: This is a prospective study evaluating the correlation between flexor tendon degeneration and the outcome of open A1 pulley release for trigger digits treatment.

I have some questions and comments regarding this study, as listed below:

In general:

- To be more specific, trigger digits refer to trigger thumb and trigger finger. Trigger fingers include the index, middle, ring, and small fingers. You can replace the term of trigger finger to trigger digits, or include the separate data of trigger fingers and trigger thumb.

Introduction:

- Line 49: The proximal part of the A2 pulley could also be a cause of trigger finger. (see: J Hand Surg Eur Vol, 2007, 32:521)

Methods:

- Line 81: It would be better to add the details of the grading of trigger finger by Green et al. in the text or a table.

- Line 85: The position setting of the JAMAR® Hand Grip Dynamometer used for testing (first, second, third...) should be mentioned. Different position settings will yield different grip strength.

- Line 94, 95: Did any of the patients undergo partial proximal A2 release? This should be clarified as the proximal A2 pulley release could be a key for some patients to achieve a complete triggering release.

Results:

- How about the correlation of tendon degeneration with the steroid injection? In addition, value to this study would be added if the outcomes of the patients with and without steroid injection could be presented and compared.

Discussion:

- Line 249, 250: “Initially, trigger finger may present as a painless clicking digit.” This is not completely true since many patients initially present with a painful digit. In addition, in the Green classification of trigger finger, grade 1 already includes “pain.”

- You included 12 digits of grade 4, indicating the trigger status in the fixed flexion contracture preoperatively. This group of patients is unique, not only their preoperative flexion contracture, but also the postoperative high chance of residual flexion contracture. You can summarize the macroscopic finding of this group of patients which is an important information. However, the functional influence by this group of patients would cause study bias which you also need to clarify.

6. PLOS authors have the option to publish the peer review history of their article (what does this mean?). If published, this will include your full peer review and any attached files.

Reviewer #1: **Yes: **Malin Zimmerman

Reviewer #2: No

---

## [Author Response · Author response to Decision Letter 0]

1 Mar 2023

Reviewer #1: The authors present an ambitious prospective study evaluating the effect of tendon degeneration on postoperative outcome following open trigger finger release.

- The manuscript requires an English revision. 

> English revision had been done.

The introduction is short and on point.

- In the methods section, please specify at what age you considered patients to be adults. 

> More than 18 years old and above. This had been included. Line 80-81.

- I would also suggest to add the specifics of the grading system by Green that you used, as not all readers will be familiar with it. 

> Green’s classification had been added. Line 101.

- Who performed the preoperative assessment and when was it performed? 

> Orthopaedics registrars assisted by a research assistant performed the assessments prior to surgery at minor operating theatre. This had been added in manuscript. Line 86.

- Did you use the English version or a translated version of the DASH?

> English version.

- When analysing the pictures, did the two surgeons do it together? Or separately? If separately, how did you handle the cases where the surgeons disagreed?

> The analysis was done by 2 surgeons separately. Training was done for both surgeons prior to assessment so they could identify and quantify the severity of tendon degeneration features (Likert scale 5) in a reliable and consistent manner. In this study, the scores obtained were averaged. Line 132-135.

- How did you define comorbidity?

> Medical conditions diagnosed by a medical professional and patients are on treatment/ follow ups.

- The authors have included several digits from the same patients, and this is a recurrent problem in hand surgical research - how to handle multiple digits from the same patients. Perhaps a small elaboration on this is the discussion would add to the manuscript - for example, it is more likely that a patient with diabetes is represented by more than one finger in the data set than a patient who is otherwise healthy.

> In this study, the trigger digit is an idiopathic condition which affects all patients with varying degree of severity and number of digits involved. Thus, analysis was done based on number of digits involved.

- I would also suggest to add a regression analysis, to be able to adjust for confounding factors. Is the seen effect of tendon degeneration on ROM and DASH a true effect, or does age and comorbidity play a role?

> The objective of this study is to investigate the association between flexor tendon degeneration and outcome of open trigger digit release while looking at the improvement of pain, ROM MCPJ/ PIPJ, pinch/ grip strength and DASH score pre- and post-surgery. The objective of this study does not require regression analysis. From previous literatures, tendinopathies may affect the outcome of trigger release. Age and comorbidity that does not involve musculoskeletal system, do not affect pain, ROM and DASH score. Comorbidity which involves musculoskeletal system like neurological disorder and inflammatory arthritis were excluded from this study.

- In the results section, I was very surprised to see that patients had on average 34 visits to the physio before surgery, and few patients had recieved corticosteroid injections beforehand. This differs quite substantially from my practice. Is it standard in your practice that patients have such an extensive treatment by physios before surgery?

> Yes. In our centre, many patients refused surgery even after counselling of its benefits. They preferred physiotherapy and were seen weekly. From our data, there were 2 patients who had 50 sessions and 8 patients who had about 40 sessions.

- I am also curious of how you reason with corticosteroids, would a previous injection affect the appearance of the tendon?

> We identified 26 trigger digits which had steroid injections. Correlation between number of steroid injections with severity of tendon degeneration was done and included in manuscript. Line 196-202 (results) and line 283-290 (discussion). Table 4 had been revised too.

- In the discussion, it is mentioned that sonographic assessment was made, however this is not specified in the methods- or is line 210-211 only referring to previous literature?

> Yes. From previous literatures, there were 2 methods of describing flexor tendon degenerations; sonographically and histologically. We did not perform these 2 methods. We described the macroscopic features of tendon degeneration via intraoperative imagings.

- Figure 4 is difficult to read and requires a higher pixel solution.

> Higher pixel of Figure 4 had been included.

Reviewer #2: This is a prospective study evaluating the correlation between flexor tendon degeneration and the outcome of open A1 pulley release for trigger digits treatment.

I have some questions and comments regarding this study, as listed below:

In general:

- To be more specific, trigger digits refer to trigger thumb and trigger finger. Trigger fingers include the index, middle, ring, and small fingers. You can replace the term of trigger finger to trigger digits, or include the separate data of trigger fingers and trigger thumb.

> The term trigger finger had been replaced with trigger digit.

Introduction:

- Line 49: The proximal part of the A2 pulley could also be a cause of trigger finger. (see: J Hand Surg Eur Vol, 2007, 32:521)

> This information had been added and the reference is included. Line 52-53.

Methods:

- Line 81: It would be better to add the details of the grading of trigger finger by Green et al. in the text or a table.

> A table had been added. Line 101.

- Line 85: The position setting of the JAMAR® Hand Grip Dynamometer used for testing (first, second, third...) should be mentioned. Different position settings will yield different grip strength.

> Description of positioning of JAMAR® Hand Grip Dynamometer had been included in manuscript. Line 94-100.

- Line 94, 95: Did any of the patients undergo partial proximal A2 release? This should be clarified as the proximal A2 pulley release could be a key for some patients to achieve a complete triggering release.

> None of our patients had proximal A2 pulley release because the triggering had completely resolved after A1 pulley release. This had been included in ‘Intraoperative Examination’. Line 121-122.

Results:

- How about the correlation of tendon degeneration with the steroid injection? In addition, value to this study would be added if the outcomes of the patients with and without steroid injection could be presented and compared.

> We identified 26 trigger digits which had steroid injections. Correlation between number of steroid injections with severity of tendon degeneration was done and included in manuscript. Line 196-202 (results) and line 283-290 (discussion). Table 4 had been revised too.

Discussion:

- Line 249, 250: “Initially, trigger finger may present as a painless clicking digit.” This is not completely true since many patients initially present with a painful digit. In addition, in the Green classification of trigger finger, grade 1 already includes “pain.”

> We had revised the statement. Line 275-277.

- You included 12 digits of grade 4, indicating the trigger status in the fixed flexion contracture preoperatively. This group of patients is unique, not only their preoperative flexion contracture, but also the postoperative high chance of residual flexion contracture. You can summarize the macroscopic finding of this group of patients which is an important information. However, the functional influence by this group of patients would cause study bias which you also need to clarify.

> We agree that grade IV trigger digit group is unique. We speculated that the flexor tendon degeneration would be severe in this group. However, there was no correlation between severity of flexor tendon degeneration with severity of trigger finger (grading). Thus, we felt there is no need for subgroup analysis.

---

## [Decision Letter · Decision Letter 1]

26 Apr 2023

PONE-D-22-30862R1Flexor tendon degeneration affects short-term outcomes of open trigger digit releasePLOS ONE

Dear Dr. Gunasagaran,

Thank you for submitting your manuscript to PLOS ONE. After careful consideration, we feel that it has merit but does not fully meet PLOS ONE’s publication criteria as it currently stands. Therefore, we invite you to submit a revised version of the manuscript that addresses the points raised during the review process. Dear Authors, one expert in the field revised your new manuscript version detecting some minor issues you should consider during the revision process.

We look forward to receiving your revised manuscript.

Kind regards,

Emiliano Cè

Academic Editor

PLOS ONE

Journal Requirements:

Reviewers' comments:

Reviewer's Responses to Questions

**Comments to the Author**

1. If the authors have adequately addressed your comments raised in a previous round of review and you feel that this manuscript is now acceptable for publication, you may indicate that here to bypass the “Comments to the Author” section, enter your conflict of interest statement in the “Confidential to Editor” section, and submit your "Accept" recommendation.

Reviewer #1: (No Response)

2. Is the manuscript technically sound, and do the data support the conclusions?

Reviewer #1: Yes

3. Has the statistical analysis been performed appropriately and rigorously? 

Reviewer #1: Yes

4. Have the authors made all data underlying the findings in their manuscript fully available?

Reviewer #1: Yes

5. Is the manuscript presented in an intelligible fashion and written in standard English?

Reviewer #1: Yes

6. Review Comments to the Author

Reviewer #1: Thank you for the revised manuscript. I only have a few small comments as follows:

Please add to the manuscript that you used the English version of the DASH questionnaire.

Please add your definition of comorbidity to the manuscript.

Interesting results with the steroid injections. I would suggest to change the heading in Table 4 from "Number of steroid injections" to "previous steroid injection".

Congratulations on a nice study!

7. PLOS authors have the option to publish the peer review history of their article (what does this mean?). If published, this will include your full peer review and any attached files.

Reviewer #1: **Yes: **Malin Zimmerman

---

## [Author Response · Author response to Decision Letter 1]

3 May 2023

Reviewer #1: Thank you for the revised manuscript. I only have a few small comments as follows:

Please add to the manuscript that you used the English version of the DASH questionnaire.

This had been added at Line 92 and 149.

Please add your definition of comorbidity to the manuscript.

This had been added at Line 184.

Interesting results with the steroid injections. I would suggest to change the heading in Table 4 from "Number of steroid injections" to "previous steroid injection".

This had been changed in Table 4.

---

## [Decision Letter · Decision Letter 2]

15 May 2023

Flexor tendon degeneration affects short-term outcomes of open trigger digit release

PONE-D-22-30862R2

Dear Dr. Gunasagaran,

We’re pleased to inform you that your manuscript has been judged scientifically suitable for publication and will be formally accepted for publication once it meets all outstanding technical requirements.

Kind regards,

Emiliano Cè

Academic Editor

PLOS ONE

Additional Editor Comments (optional):

Reviewers' comments:

Reviewer's Responses to Questions

**Comments to the Author**

1. If the authors have adequately addressed your comments raised in a previous round of review and you feel that this manuscript is now acceptable for publication, you may indicate that here to bypass the “Comments to the Author” section, enter your conflict of interest statement in the “Confidential to Editor” section, and submit your "Accept" recommendation.

Reviewer #1: All comments have been addressed

2. Is the manuscript technically sound, and do the data support the conclusions?

Reviewer #1: (No Response)

3. Has the statistical analysis been performed appropriately and rigorously? 

Reviewer #1: (No Response)

4. Have the authors made all data underlying the findings in their manuscript fully available?

Reviewer #1: (No Response)

5. Is the manuscript presented in an intelligible fashion and written in standard English?

Reviewer #1: (No Response)

6. Review Comments to the Author

Reviewer #1: (No Response)

7. PLOS authors have the option to publish the peer review history of their article (what does this mean?). If published, this will include your full peer review and any attached files.

Reviewer #1: **Yes: **Malin Zimmerman

---

## [Editor Report · Acceptance letter]

18 May 2023

PONE-D-22-30862R2 

Flexor tendon degeneration affects short-term outcomes of open trigger digit  release 

Dear Dr. Gunasagaran:

I'm pleased to inform you that your manuscript has been deemed suitable for publication in PLOS ONE. Congratulations! Your manuscript is now with our production department. 

Kind regards, 

on behalf of

Professor Emiliano Cè 

Academic Editor

PLOS ONE